# Peptide Toxins from Antarctica: The Nemertean Predator and Scavenger *Parborlasia corrugatus* (McIntosh, 1876)

**DOI:** 10.3390/toxins16050209

**Published:** 2024-04-30

**Authors:** Erik Jacobsson, Adam A. Strömstedt, Håkan S. Andersson, Conxita Avila, Ulf Göransson

**Affiliations:** 1Pharmacognosy, Department of Pharmaceutical Biosciences, Uppsala University, Box 591, 75123 Uppsala, Sweden; erik.jacobsson@uu.se (E.J.); adam.stromstedt@uu.se (A.A.S.); 2Department of Medical Biochemistry and Biophysics, Karolinska Institutet, 17177 Stockholm, Sweden; hakan.andersson@ki.se; 3Department of Evolutionary Biology, and Ecology, Environmental Sciences, Biodiversity Research Institute (IrBIO), Faculty of Biology, University of Barcelona, Av. Diagonal 643, 08028 Barcelona, Spain; conxita.avila@ub.edu

**Keywords:** nemertean worm *Parborlasia corrugatus*, peptide toxin, *Artemia* assay, membrane permeabilization assay

## Abstract

Peptide toxins from marine invertebrates have found use as drugs and in biotechnological applications. Many marine habitats, however, remain underexplored for natural products, and the Southern Ocean is among them. Here, we report toxins from one of the top predators in Antarctic waters: the nemertean worm *Parborlasia corrugatus* (McIntosh, 1876). Transcriptome mining revealed a total of ten putative toxins with a cysteine pattern similar to that of alpha nemertides, four nemertide-beta-type sequences, and two novel full-length parborlysins. Nemertean worms express toxins in the epidermal mucus. Here, the expression was determined by liquid chromatography combined with mass spectrometry. The findings include a new type of nemertide, 8750 Da, containing eight cysteines. In addition, we report the presence of six cysteine-containing peptides. The toxicity of tissue extracts and mucus fractions was tested in an *Artemia* assay. Notably, significant activity was observed both in tissue and the high-molecular-weight mucus fraction, as well as in a parborlysin fraction. Membrane permeabilization experiments display the membranolytic activity of some peptides, most prominently the parborlysin fraction, with an estimated EC_50_ of 70 nM.

## 1. Introduction

Investigations into the peptide and protein toxins from marine organisms have yielded a range of fascinating compounds, including conotoxins [1] and cnidarian toxins [2]. These toxins exhibit a wide variety of activities, including ion channel activity [3], insulin-like effects [4], and membranolysis [5]. Some of these peptide toxins are of great interest for drug development, as exemplified by Prialt^®^ (ziconotide), which is already on the market for chronic pain.

Many of the studies aimed at finding new toxins have been focused on a few taxa, ignoring certain less-studied groups of venomous animals entirely [6]. One relatively neglected source of peptide and protein toxins is the nemertean worms, especially those found in remote areas of marine environments. Here, we investigate such a worm from a remote area—the Antarctic *Parborlasia corrugatus*.

*P. corrugatus* is a Cerebratulidae nemertean worm, also called ribbon worms because they are flat, long, and smooth. This group of animals is characterised by the use of a proboscis both to capture food and to defend themselves. Being a voracious top predator and scavenger of the Antarctic ocean floor, *P. corrugatus* may grow to a length of approximately three metres with a diameter of two to five centimetres. *P. corrugatus* is common in the Southern Ocean, often found in feeding groups, and has been reported in shallow waters (5–15 m) down to approximately 3500 m below sea level [7]. In spite of living openly on the ocean floor and appearing to be attractive prey, *P. corrugatus* is not palatable to known predators in its environment. This is probably due to the molecules secreted within its mucus. Indeed, two Antarctic fish species, *Dissostichus mawsoni* and *Trematomus bentacchii*, have been found to reject pieces of *P. corrugatus* tissue in feeding experiments [8], while the macropredator sea star *Odontaster validus* did not clearly reject *P. corrugatus* lipophilic extracts imbedded into shrimp pieces [9,10]. The lipophilic extract of *P. corrugatus* was also toxic against the sperm and early development stages of the sympatric sea urchin *Sterechinus neumayeri* Meissner [11].

In addition to the acrid taste, the mucus also harbours a small group of cytolysins that are around 10 kDa proteins known to lyse erythrocytes, the so-called parborlysins [12]. In total, seven parborlysin isoforms have been revealed by PCR, using primers for DNA sequences corresponding to the parborlysin already discovered [13]. Cytolysins with high sequence similarity to parborlysins have also been found in other nemertean worms [14] along with large neurotoxic peptides (B-neurotoxins/beta-nemertides; 6 kDa, four disulphides) [15,16] and smaller cystine-knot-containing peptide toxins (the ~3 kDa alpha nemertides) [16].

Small-molecule toxins have also been reported from ribbon worms, most notably the famous Fugu toxin tetrodotoxin, TTX [17,18], although recent work puts some of these observations into question [19,20]. TTX is, however, unquestionably present in other species as determined by NMR [18]. Another interesting compound found mostly in the anterior part of the proboscis in some hoplonemertean worms is anabaseine [21]. Anabaseine and derivatives thereof have been investigated for their possible use as Alzheimer’s disease medication [22]. The toxins from nemertean worms have been thoroughly reviewed by Göransson et al. [23].

Here, we investigate the peptide and small-protein toxin content in *P. corrugatus* using transcriptomics, and liquid chromatography coupled to mass spectroscopy is used to investigate the mucus at the peptide level. Initial bioactivity is explored in an *Artemia salina* microwell assay and membranolytic activity is investigated for selected toxins in a liposome permeabilization assay.

## 2. Results

### 2.1. Transcriptome

The Illumina 2500-sequenced and Trinity-assembled transcriptome produced in this study resulted in 67,148 contigs with a combined contig length of 43,298,283 bases. The getorf (−minsize 70) translated database yielded 968,918 possible ORFs, with 38,537 sequences longer than 100 amino acids.

A publicly available transcriptome (NCBI id: SRX731467), previously utilised in phylogenetic studies [24,25], was assembled using the same protocol, resulting in 8174 contigs, with a total contig length of 2,298,549 bases. The translated database (getorf, −minsize 70) resulted in 64,719 possible ORFs and a total of 2,639,313 amino acids (aas). However, only 1393 of the ORFs are equal to or longer than 100 amino acids.

For the purpose of peptide discovery, judged by the number of long ORFs that enable the discovery of full-length peptide toxins and/or precursors, the quality of the transcriptome reported here is superior. Sequence identifiers in the Section 2 that start with c originate from the transcriptome sequenced in this study, while identifiers starting with DN were found in the already published transcriptome.

### 2.2. Transcriptome Mining Revealed Parborlysin-Like Sequences

The combined BLAST and MAST strategy, using previously described parborlysins as queries, revealed 12 sequences with similarity to the parborlysins reported by Berne et al. [12]. However, none of the sequences were found to be identical to previously reported parborlysins. The conoprec tool only predicted the correct pre-toxin cleavage site for mature toxins compared to the reference sequences. All sequences were found to be highly similar, with few substitutions apart from truncations. Furthermore, conoprec did not predict correct post-toxin cleavage sites of all seven reported parborlysins. Instead, the sequences were processed using SignalP-5.0 [26], which does not divide the sequences further into pre- and post-sequences. The final ClustalW alignment is displayed in Figure 1.

### 2.3. Identification of Nemertide Beta and Neurotoxin-B-like Sequences

BLAST and MAST searches with nemertide beta 1 and neurotoxins of the B type (B-II and B-IV) as queries resulted in a total of four hits after alignment and manual inspection. The conoprec-predicted cleavage of pre-sequences is in accordance with the reported sequence boundaries for nemertide beta 1 and neurotoxins B-II and B-IV. Two of the hits (named: beta_c38679 and beta_DN6412) have a loop4 length between Cys4 and Cys5 of 10 amino acids. The corresponding length of the remaining two hits, beta_DN6109 and beta_c52422, is 16 amino acids (Figure 2A).

As described below, one peptide could be partially sequenced from the epidermal mucus of *P. corrugatus*. When using a partial sequence of that peptide, called “8750”, to BLASTp query the transcriptomes, the full precursor sequence and one additional hit was identified, DN6723. Both sequences were subjected to conoprec processing, resulting in accurate prediction of the mature 8750 Da peptide. The DN6723 sequence is identical to 8750 in positions 14–82. It does not, however, contain a distinguishable signal sequence and is probably truncated in the C-terminal (Figure 2B).

### 2.4. Parborlasia Corrugatus Expresses Alpha Nemertides

The nemertide alpha family of seven full-length toxins was screened against the *P. corrugatus* transcriptomes using the MEME suite and BLASTp. The resulting hits were processed using conoprec and the mature sequences were filtered for duplicates and aligned using ClustalW. Masses for the predicted mature presumed toxins were calculated, and were found to range between 3266 and 4720 Da. The numbers of amino acids in loops between cysteines vary in the found sequences, especially in loops 3 (see 4057Da_c38595) and 4 (Figure 3A).

Again, a partial sequence could be identified at the peptide level using tandem mass spectrometry (MSMS). A sequenced peptide with a mass of 3624 Da was found to harbour six cystines participating in three disulphides. Fuzzpro hits in the de novo MS-sequenced fragment were collected and used as queries in BLASTp searches in the transcriptomes. The BLASTp revealed a total of five sequences with a high degree of similarity to the identified sequence (Figure 3B). Interestingly, one of the hits (3266Da_c10532) was also found in the nemertide alpha hit list. Full precursors of hits from both 3625 and alpha surveys (3649Da_913_c2426, 3716Da_744_c37678 and 3266Da_c10532) are displayed together with the reference nemertide alpha 1 in Figure 3C.

### 2.5. BLASTp of NCBI nr Reveals No or Few Homologues of Identified Toxins

The combined and manually evaluated and collected sequences from the different types of nemertides (Alpha, 3716, 3624, 8570, beta/neurotoxin-B, and parborlysin/cytolysin-A) were blasted against the NCBI non-redundant (nr) database using BLASTp. For the sequences similar to alpha nemertides, no hits with high similarity were found, except for alpha 1 and 2. The results for the identified beta/neurotoxin-B hits were surprisingly not identified back to the Neurotoxin B-II and B-IV.

No hits with sequence similarity to 3624 were found in the NCBI non-redundant protein database. The closest hit to the identified 8750 Da peptide was found to be a predicted uncategorized protein from *Hydra vulgaris* (XP_012564781.1) with an E-value of 0.006. The *H. vulgaris* sequence shares the cysteine framework with the 8750 Da peptide. In the group of parborlysins, no significant hits outside the members of published parborlysins/cytolysin A were identified.

### 2.6. Functional Prediction of BLASTp Hits Using CSPred

The online prediction server CSPred [27], optimised for cystine-stabilised peptides, was used to predict the probability of biological functions in five functionality classes: ion channel blocker, anti-microbial peptides (AMPs), acetylcholine receptor inhibitors, serine protease inhibitors and haemolytic activity.

The CSPred probability prediction data for the parborlysin/cytolysin-A group of peptides show high (94–100%) probability for AMP classification, including the reference parborlysins 1–7 and cytolysin A-III. The related functionality, haemolysis, was predicted in a wide range of probabilities: 0.01–74%, with the reference Cytolysin A-III at 74%. The previously described parborlysins 1–7 were predicted to belong to the haemolytic class with probabilities of 34 (parborlysin 1) to 71% (parborlysin 2). None of the sequences were predicted to belong to the other classes (probability lower than 9%).

The 8750 Da peptide was predicted to have an 83% chance of belonging to the AMPs, and a low probability for the other functionality groups (<6%). The sequence with similarity to 8750, DN6723, displays a 56% similarity to the AMP pattern in the CSPred prediction, and no more than 11% probability of belonging to any other classes.

The CSpred probability predictions for the sequences similar to beta/neurotoxin B-IV resulted in low scores for ion channel activity: c38679 (65%), c52422 (56%), and DN6412 (50%). In contrast, the Neurotoxin B-II and B-IV scored above 98%, while beta 1 scored 59% in the ion channel activity probability prediction. All peptides including the references scored a probability below 30% in the remaining classes of targets.

The BLASTp and mast hits for the alpha nemertide toxins scored high in the probability of ion channel activity. The reference sequences, alpha 1–7, were all found to have a score in the range of 87 (alpha 7) to 92% (alphas 1, 3, and 5). Four of the transcriptome hits (3266Da_c10532, 4110Da_c31103, 4720Da_c41739, and 4281Da_c45383) scored 87–97% ion channel activity probability, while the remaining two sequences, 4057Da_c38595 and 4281Da_c45383, scored 64 and 34%, respectively. The alpha hits also had a probability of AMP activity with a range of 3–59%.

The sequences with similarity to the 3624 Da peptide display a spread in probability prediction; 3649Da_913_c2426 had a 57% probability of ion channel activity and a 42% probability of serine protease inhibitor activity. The closely related sequence 3716Da_744_c37678 scored highest in serine protease inhibitor activity, with 44%. Another sequence, 3979Da_DN758, had an 89% probability of belonging to the AMP class of peptides, 51% probability of ion channel activity, and 38% probability of serine protease inhibitor activity.

### 2.7. Effect of Crude Extract and Parborlysin Fraction on Artemia salina

Five samples were tested in an *Artemia* microwell assay: (1) body extract; (2) lyophilised crude mucus-containing water; (3) a low-molecular-weight (LMW) fraction; (4) a high-molecular-weight fraction (HMW); and (5) a fraction of mixed parborlysins. No effect could be seen, at the concentrations tested, for the crude mucus-containing water and LMW samples. The body extract killed all test organisms at the highest concentration employed, 10 mg/mL (280 nm: 6.0 mg/mL), with 6.25% at 1 mg/mL (280 nm: 0.61 mg/mL). In the HMW experiments, all *A. salina* were killed at a concentration of 0.28 mg/mL (280 nm: 0.18 mg/mL). No toxicity could be observed at lower concentrations. For the sample containing a mix of parborlysins, 100% toxicity was found at 1.0 mg/mL and 7.1% toxicity at a concentration of 0.1 mg/mL, whereas no toxicity could be observed at lower concentrations (Table 1).

### 2.8. LCMS Analyses of Mucus Reveal Complex Peptide Expression

A UPLC-QToF profile of re-suspended mucus revealed a plethora of peaks in the mass range between 300 and 2000 m/z. Deconvoluted masses reveal peptides with m/z that may, at a glance, be roughly divided into size classes of 3.5–5 kDa and probable parborlysins at 9–10.5 kDa. The 9–10.5 kDa proteins tend to elute later than the smaller class. Furthermore, the 9–10.5 kDa compounds elute close to each other in this system. Despite repeated attempts, with, e.g., lower slope gradients, we were unable to separate the compounds. Furthermore, the reduction and alkylation of an aliquot followed by PD-10 gel filtration revealed four peptides containing six Cys residues forming three disulphide bonds, and one peptide (8750 Da) with eight cystines involved in four disulphide bonds (Figure 4).

### 2.9. Isolation and Sequencing of 8750 and 3625

Two peptides were isolated and sequenced using UPLC-QToF. These peptides have deconvoluted masses of 8750 and 3624. The 8750 peptide was found to increase in mass by 464 u after iodoacetamide alkylation, corresponding to eight Cys residues involved in four disulphide bonds (Figure 5).

The mass difference between the native and alkylated 3624 peptide was found to be 348 Da, corresponding to six cystines forming three disulphides. Another peak at 21 min, with a deconvoluted mass of 3957 (+332 Da), was also identified in the reduced and alkylated sample. The difference between the six-Cys alkylated peptide 3973 and the 3957 peak is 16 Da (Figure 6).

The 8-Cys 8751 peptide was found to be cleaved into two main fragments following trypsinisation; 1101.78^1+^ and a larger fragment, 1240.77^4+^. The sequence of the 1101 fragment was found to correspond to SPANV[I/L]WTT[I/L] (positions 75–84) by *de novo* sequencing, as exemplified in Appendix A. The sequence was used as a fuzzpro query in the transcriptomes. The larger fragment, 1240^4+^, displays a complex MSMS spectrum. It does, however, correspond in total mass to positions 20–65 in the sequence found using fuzzpro.

Glu-C cleavage yielded a fragment, 1174.29^2+^, with a deducible sequence corresponding to positions 21 to 43 (ATGQTAGYVS[I/L]VGCCPSGYGLE). In the chymotryptic cleavage, two ions were identified, corresponding to positions 45–51 [I/L]DGVGAF (678.34 1 z), and the miscleaved 41–51: GLEY[I/L]DGVGAF (1140.56^1+^). The combined data for the 8751 peptide correspond to positions 41–65 and the C-terminal positions 75–84, i.e., 56 of 84 (67%) amino acids sequenced at the peptide level (Figure 7A).

Trypsinisation of the reduced and alkylated 3625 peptide revealed one fragment peak at 1174^2+^ and 783^3+^, from which a stretch of sequence was deducible from b and y ions in tandem-MS experiments: 577-THACF[I/L]NSNCCQQPR. The sequence stretch was used as query for the transcriptomes and a hit with similar, but not identical, sequence was identified. The Xle was determined to be Leu from homology to the sequence from genetic data (Figure 7B).

### 2.10. Parborlysins Have Potent Liposome-Disrupting Activity

Two peptides were isolated by HPLC-UV in sufficient amounts and with purity for bacterial membrane permeabilization characterisation: the 3624 Da and 8750 Da peptides. In addition, a fraction of parborlysin-sized (9.5–10.5 kDa) small proteins were collected.

The ability of the *Parborlasia* peptides to permeabilize reconstituted *E. coli* phospholipid membranes was tested in a liposome-based assay. The peptides display the typical steep logarithmic AMP leakage kinetic (Figure 8) [28]. The fraction containing parborlysins was found to be highly active using a generic bacterial model membrane type with an EC_50_ of 0.07 µM. The 3624 Da and 8750 Da peptides display EC_50_ values of 0.9 and 1.8 µM, respectively. The 3624 Da peptide was tested in a single experiment series in this assay. The 26-residue long bee venom peptide melittin was used as a reference compound and exhibited permeabilization at an EC_50_ of 0.3 µM (Figure 8). Previous experiments have attained an EC_50_ of 0.4 µM for melittin [28], the error margin likely being a consequence of the steep leakage potency curve and the absence of markers close to the EC_50_ in our experiment. The nemertean peptide EC_50_ data, on the other hand, do not suffer from this quality issue.

Since melittin is one of the most potent membrane lytic peptides known, the results indicate that the parborlysins in particular, but also the 3624 Da and the 8750 Da peptides, are potent inducers of liposome leakage.

## 3. Discussion

Here, we describe the characterisation of nemertea-related peptide and protein toxins on the transcriptomic level in *P. corrugatus*. We also report the discovery of a new type of small protein from the excreted mucus. The protein, 8750 Da, was partly sequenced by tandem MS and the full sequence was identified from transcriptomic data. The sequence shows similarity to, and shares cystine motif with, an uncharacterised protein from the freshwater hydrozoan *Hydra vulgaris* Pallas, 1766. The cysteine motif has not been reported in the specialised conotoxin and arachnoid databases conoserver [29] and arachnoserver [30].

In addition to the 8750 Da peptide, transcriptome mining using BLASTp and the MEME suite revealed sequences similar to all types of reported nemertides: parborlysin/cytolysin-A, beta/neurotoxin-B, and alpha nemertides.

In total, three full-length sequences of parborlysins were identified in the transcriptomic data. However, none of the previously reported parborlysins/cytolysins-A [13,14] could be identified in the two available transcriptomes. In addition to the full-length sequences, several truncated hits were identified, sharing the cysteine pattern with the reference sequences. The conoprec tool [29], optimised for conotoxins, was found to not predict the mature toxins in a sufficient manner. Prediction of possible C-terminal cleavage is harder to assess; only the N-terminal sequence was determined in the original parborlysin paper [12], and the masses of seven sequences reported later [13] do not correspond to the MS spectrum reported by Berne et al. [12]. The discrepancy in mass sequence relationship could possibly indicate the presence of post-translational modifications.

Four nemertide beta/neurotoxin-B sequences were identified at the RNA level, two of which contain the extended loop4, also found in beta nemertides, while the remaining two sequences lack the loop4 extension and hence are more similar to neurotoxin B II and IV.

The sequences with similarity to alpha nemertides, seven in total, may be divided into three groups based on the difference in length of amino acid stretches in loops 3 and 4. One group contains similar lengths in all loops compared to alpha nemertides, while one sequence (4057Da_c38595) has an extended loop 3. Three sequences (3974Da_pc_2_c0, 3335Da_c55849, and 4281Da_cc45383) show extension in loop 4. The number of nemertide alpha-like sequences found in the transcriptome is relatively high compared to other nemertean transcriptomes previously investigated. The highest number of alpha nemertides found in any nemertean transcriptome is three [16]. Even though the 3624 peptide sequence could not be fully determined by tandem MS experiments, a large part of the sequence could be used to find presumably similar sequences in the transcriptomes. The sequences with high similarity share a cysteine pattern with alpha nemertides. Also, the full precursor sequences show similarity.

The tandem MS sequencing of 3624 Da of the major fragment in the tryptic cleavage, with ions 1174^2+^ and 783^3+^, could be partly elucidated in positive MS mode. No ion for the C-terminal sequence could be observed. This is not surprising if the peptide is highly similar to the 3649 Da and 3716 Da peptides: the C-terminal sequence of these peptides are highly acidic, and cannot be anticipated to give rise to a high abundance of ions in positive MS mode.

To identify disulphide-containing peptides and proteins in the excreted mucus, an aliquot of mucus was reduced and alkylated and compared to its native counterpart. The iodoacetamide alkylation of cystines involved in disulphides gives rise to a mass increase of 58 Da per Cys residue. When the two mass chromatograms were compared, we could identify four peptide masses that had shifted by 348 Da, i.e., six cystines forming disulphides: 4231, 3718, 3625, and 3975. One peptide, the 8750 Da peptide, was found to have a mass increase corresponding to eight cystines (464 Da). Interestingly, the 3718 Da peptide (c3718 in Figure 4) corresponds to the contig 3716Da_744_c37678 sequence by mass (average 1 z vs. predicted monoisotopic) and number of Cys residues. Moreover, the shift in retention to a more hydrophilic region in the chromatogram after alkylation is indicative of a native constrained structure with hydrophobic residues forced to the surface, i.e., as with alpha nemertides [31] and the 3624 Da peptide, indicating similar structural features.

The toxicity to *A. salina* of five different sample types—(1) crude body extract; (2) mucus-containing water; (3) LMW mucus; (4) HMW mucus; and (5) a fraction containing a mix of parborlysins—was tested. The mucus-containing water and the LMW fraction probably both contain mostly salts from the seawater and exhibited no toxicity at the concentrations tested. The parborlysin fraction exhibited the highest activity in the *Artemia* assay; if the mixture of parborlysins is considered to have an average molecular weight of 10 kDa, 1.0 mg/mL would correspond to approximately 100 µM. The neurotoxic alpha nemertide toxins exhibited EC_50_ values in the same assay at 0.3–5.2 µM as a comparison [31].

The relatively low toxicity of the body extract in comparison with the HMW and parborlysin toxicities indicates a defence function of at least the larger compounds in the mucus of *P. corrugatus,* as described for other nemertean worms [16,23]. However, none of the peptides with sequence similarity to alpha and beta nemertides from *P. corrugatus* have been tested for neurotoxicity, or even general toxicity to any organism so far. Injection of crude lyophilised mucus extracts from *P. corrugatus* into crayfish did not, however, cause paralysis at 10 mg/kg doses [12].

We could not separate the parborlysin isoforms through HPLC (C18) in any gradient mode tried. The problem is illustrated in Figure 4, in which UPLC was used for separation with samples in minute amounts and a slow gradient. Berne et al. were also unable to separate the parborlysins despite exhaustive attempts using both C18 and cation exchange resins [12]. Hence, we were forced to use a parborlysin fraction in the assays rather than any pure compound.

The membrane permeabilization assay shows that parborlysins are highly membranolytic compounds, displaying typical AMP leakage kinetics (Figure 8). This is in agreement with the CSPred prediction, which classified parborlysins into the AMP category of cystine stabilised peptides as well as classifications into other membranolytic peptide categories. In molar equivalents, they are almost five times more potent than the potent benchmark peptide, melittin. They are also 11 times more active than LL-37 on identical bacterial membranes [28].

Permeabilization on *E. coli* liposomes has a strong correspondence with the reciprocal bactericidal effect on *E. coli* bacteria as well as bacteria in general [28]. Melittin-level activity on bacterial liposomes is also typically associated with membranolytic activity on other cell types, including erythrocytes, as the mechanism generally has somewhat narrow target selectivity that is not governed by specific affinities [28].

In order to reach the parborlysin level of potency, high membrane binding affinity is a prerequisite. For highly anionic membranes such as the *E. coli* liposomes and corresponding bacteria, this involves not only hydrophobic interaction between peptides and the membrane interior but also electrostatic interactions with the anionic membrane headgroup area region. With few exceptions, AMPs are highly cationic and so are the parborlysins identified here; their net positive charge is between 16 and 19 (Table 2). A noteworthy difference is the increment of the leakage-to-concentration curves (Figure 8). Although they are all sigmoidal in nature, those of the nemertean peptides are markedly less steep than that of melittin. The coefficient of the nemertean peptides at EC50 is similar to that of cyclotides, while melittin is similar to LL-37 [28]. Melittin and LL-37 are both linear alpha-helical peptides, with their hydrophobic patch along one side of the helix. Peptides of this configuration can go from horizontal to transmembrane orientation (in the case of melittin, a pseudotransmembrane alignment) when reaching a critical membrane concentration, which is associated with leakage [32]. Parborlysins have been suggested to consist mainly of alpha helices restricted to a core by three disulphides [13]. It is possible that the similarities in leakage-to-concentration coefficients are linked to this peptide property.

These results, together with previously demonstrated haemolytic activity of a heterogenous parborlysin sample to bovine erythrocytes at an HC_50_ of 5.75 µM [12], indicate a general membranolytic function.

The 3625 Da and 8750 Da peptides were also tested for membrane disruptive activity on the bacterial liposomes, displaying substantial permeabilization properties, with EC_50_ values of 0.9 and 1.8 µM, respectively. Both of these values are low enough to indicate that membrane permeabilization is a function for these compounds. The cocktail of bioactive proteins and peptides found in *P. corrugatus* thus constitute a cocktail of membrane-disruptive peptides, and exactly these types of complex yet redundant compositions of host AMPs with similar function seem to be a reoccurring feature in nature with regard to animal defence. This is a powerful explanation for how these animals survive with their soft muscular bodies in Antarctic ecosystems under strong predatory pressure.

## 4. Conclusions

Here, we present a suite of predicted nemertides found at the transcriptomic level from the top Antarctic predator, the nemertean worm *P. corrugatus*. Peptides and proteins were also mapped at the protein level, and the findings include a new type of nemertide in the epidermal mucus with a unique cysteine pattern. The nemertide has a mass of 8750 Da and contains 84 amino acids, of which eight are cysteines involved in four disulphide bonds. We also report two new full-length and several partial parborlysin/cytolysin sequences. In addition, we found four nemertide beta/neurotoxin-B-like contigs, and a total of 10 sequences with a cysteine pattern similar to that of alpha nemertides. Furthermore, we provide toxicity data for a small collection of extracts and fractions, including a full body extract, low- and high-molecular-weight fractions of mucus, and a fraction of parborlysins. Finally, the study of the membrane permeabilization properties of the nemertean peptides find them all to be involved in membrane-disruptive mechanisms. Most prominent were the parborlysins that reached an EC_50_ as low as 70 nM on bacterial liposomes, 5-fold more potent than melittin. A future full structural characterisation of parborlysins as well as the 8750 mass nemertide is likely to provide further insights into the action and potential utility of these toxins. Antarctic organisms continue to be amazing sources of new molecules yet to be fully studied.

## 5. Materials and Methods

### 5.1. Collection of Parborlasia corrugatus

*P. corrugatus* was collected during the DISTANCOM project from Antarctica, more precisely at Deception Island (LAT 62° 58.18′ S, LONG 60° 42.23′ W; South Shetland Is.) in February 2016. Samples were collected by scuba diving at 5–15 m depth and transported to the lab in individual bags. Mucus was collected by agitation of specimens in beakers to release mucus into the surrounding water. Mucus samples and living specimens were quickly frozen at −20 °C. Small tissue samples were collected and stored in RNAlater (Thermo Fisher, Waltham, MA, USA) solution before freezing, according to the manufacturer’s instructions. All frozen samples and RNAlater samples were transported to the University of Barcelona (Catalonia), sorted, and further sent to Uppsala (Sweden), where mucus-containing water was lyophilised upon arrival.

### 5.2. Transcriptome Sequencing and Mining

A tissue piece of *P. corrugatus* stored in RNAlater was sent to Macrogen, Seoul, Republic of Korea, for RNA extraction and RNA sequencing (Illumina HiSeq 2500, San Diego, CA, USA). The assembled transcriptome was searched for the previously reported alpha and beta nemertides [16], neurotoxin B-II and B-IV [15], parborlysins, and cytolysin A-III [13,33] using NCBI BLAST+ (v.2.8.1 tBLASTn, BLASTp; default settings; e-value cutoff: 10).

The publicly available transcriptome of *P. corrugatus* (NCBI accession: SRX731467) [24,25] was assembled using the open-source usegalaxy.org [34] server’s implementation of Trinity, using default settings. The assembly was investigated for peptide/protein toxins as described above.

The MEME suite (standalone, v5.0.4) [35] was used to create two MEME profiles for each set of known nemertean toxin precursors: alpha, beta/B, and parborlysin/cytolysin A. The first profile corresponds to the highly conserved toxin sequences, and the second is derived from the ER signal sequence for each toxin family. MAST (v5.0.4) was subsequently used to investigate the combined *P. corrugatus* transcriptomes for the MEME profiles. The combined BLAST, fuzzpro, and MEME/MAST results for each type of peptide/protein were combined, filtered for duplicates, and manually inspected after ClustalW alignment.

Full-length precursors were analysed using the conoprec tool available through conoserver [36] for the probable identification of signal, pro-sequence, mature, and post-toxin cleavage sites. In cases where conoprec failed to predict correct cleavage sites, as judged by reference sequences, SignalP-5.0 was used [26].

Peptide sequences obtained through tandem MS experiments were searched in a getorf (default settings except: −minsize 70) translated database using the prosite pattern-based fuzzpro programme [37].

Predicted mature and precursor sequences for the different types of nemertides were blasted against the non-redundant protein sequences (nr) using BLASTp at NCBI. Furthermore, the sequences were also submitted to the CSPred webserver for preliminary probability prediction into different functionality classes [27]: ion channel blocker, AMP, acetylcholine receptor inhibitor, serine protease inhibitor, and haemolytic activity.

### 5.3. Extraction and Fractionation of P. corrugatus Tissues and Mucus

A piece of *P. corrugatus* was lyophilised (6.3 g) and pulverised using a pestle and mortar. The resulting powder was extracted in 60% AcN, filtered, and lyophilised (1.7 g, 27% yield). A dilution series in MQ water was prepared in the concentration range from 0.020 to 20 mg/mL for the *Artemia* microwell assay.

Lyophilised mucus-containing water was dissolved in 30% AcN and fractionated into an HMW fraction and an LMW fraction using size exclusion chromatography (SEC; PD-10 cartridge, GE-Healthcare), according to the manufacturer’s instructions. Both fractions were freeze-dried. A dilution series for the *Artemia* microwell assay was prepared in MQ water for each fraction with the concentration span (by weight): LMW: 0.043 to 43 mg/mL; and HMW: 0.056 to 5.6 mg/mL.

The solutions with the highest amount of extract/fraction were analysed using nanodrop spectroscopy at 280 nm (in triplicate) to crudely determine protein presence and give an indication of concentration under the premise that ‘1 Abs_280nm_ = 1 mg/mL’ (mg/mL): body extract: 12.1; lyophilised mucus: 0.20; LMW: 0.17; and HMW: 3.56.

### 5.4. Artemia Bioassay

The *Artemia* assay was performed as described by Solis et al. [38]: *Artemia salina* cysts (Ocean Nutrition^TM^, approx. 1 g/L water, Newark, CA, USA) were hatched, under constant light, in a separation funnel containing 33 g salt/L (Red Sea^TM^; coral pro salt, Herzliya, Israel) in deionised water. A small aquarium air pump was used to aerate the water until the shrimps were harvested, i.e., after 24 h.

Each lyophilised fraction/extract was dissolved in MQ water; 100 µL of the solutions was transferred to 96-well flat-bottom plates (polystyrene, untreated, Thermo Fisher 96 F untreated straight w/lid 260895). Five to fifteen *Artemia* nauplii in 100 µL of artificial sea salt water were transferred to the extract/fractions and incubated at RT for 24 h. The dead and non-moving nauplii were counted using a dissecting microscope. To count the total number of nauplii in each well, 100 µL of MeOH was added and allowed to incubate for 15 min. Toxicity was assessed by calculating the “stagnant/total nauplii” in each well; 100 µL of MQ water was used as the control. Experiments were performed in duplicate.

### 5.5. Peptide and Protein Mapping of the Mucus

Isolation of individual peptides from re-suspended and filtered mucus was performed by gradient elution RP-HPLC-UV on a semi-preparative scale CSH C-18 column (Waters, XSelect CSH C18 ODB, 10 × 250 mm, 10 µm, 130 Å), coupled to a Shimadzu 20A series system (20-AD), with UV detection (SPD 20A) at 215 and 280 nm, operated under LabSolutions (v5.52, Shimadzu). The column oven (CTO-20AC) was set to 30 °C. The gradient was set to increase from 5 to 60% AcN, 0.05% TFA, in 55 min at a flow rate of 3 mL/min, followed by a wash (97% AcN, 0.05% TFA). Fractions were collected manually upon observation of a rise in UV absorption at 215 nm. Fractions were lyophilised prior to further analysis.

### 5.6. Reduction and Alkylation of Cystines

The reduction and alkylation of a high-molecular-weight fraction from SEC, as well as isolated peptides, were performed as by Burman et al. [39]. In brief, the reduction of disulphide bridges was accomplished by the addition of dithiothreitol (DTT) in 0.25 M Tris-HCl, 6 M guanidine, and 1 mM EDTA (pH 8.5) followed by incubation under nitrogen at 37 °C for 3 h in the dark. The reduced cysteines were subsequently alkylated using iodoacetamide (IAM) by adding IAM in 0.5 M Tris-HCl and 2 mM EDTA for 3–10 min. Alkylation was quenched by the addition of 0.5 M citric acid. Upon IAM alkylation, each cystine involved in a disulphide bridge yielded a mass increase of 58 Da. Sequencing-grade trypsin, chymotrypsin, and Glu-C (Promega, Madison, WI, USA), in different experiments, were used to cleave alkylated peptides into smaller pieces suitable for tandem MS sequencing [40].

### 5.7. Mass Spectrometry

UPLC-ESI-QToF analyses were performed on a Waters nanoAquity system coupled to a Waters Micromass Qtof micro mass spectrometer, all operated under the MassLynx v4.1 software. The column used was an ACQUITY UPLC M-Class Peptide BEH C-18 (75 µm × 250 mm, 300 Å) with a gradient from 1% AcN, 0.1% FA to 90%, 0.1% FA in 50 or 180 min, at a flow rate of 0.3 µL/min. Detection was performed in positive electrospray mode at 4300 V, with a detection span of 300–2000 *m*/*z* for single MS. For tandem MS, the survey scan was set to 300–2000 *m*/*z* and MS/MS scan 50–2000 *m*/*z*.

### 5.8. Bacterial Membrane Permeabilization Assay

Liposomes were manufactured and permeabilization-assayed as described previously [39]. Briefly, lipid films of *Escherichia coli* polar lipid extract were deposited on round-bottom flask walls. The lipid films were re-suspended at 55 °C in an aqueous solution of 100 mM of 5(6)-carboxyfluorescein in Tris buffer. The suspensions were subjected to repeated extrusion through a 100 nm polycarbonate membrane in order to reduce multilamellar structures and polydispersity. Un-trapped carboxyfluorescein was removed by gel filtration. Membrane permeability was measured by monitoring carboxyfluorescein efflux from the liposomes to the external low-concentration environment, resulting in a loss of self-quenching and an increased fluorescence signal. The 96-well plates were prepared with a 2-fold serial dilution of the peptides in Tris buffer (10 mM, pH 7.4), and controls without peptides (background) and 0.16% Triton X-100 (maximum leakage). The plates were administered a liposome solution to a final lipid concentration of 10 μM in 200 µL. The effects of each peptide concentration on the liposomes were monitored for 45 min, at which point the initial leakage had largely subsided. The results shown represent the mean from four experiments (one for 3624 Da) with standard deviations and are expressed as the percent of total leakage generated with Triton X-100 and subtraction of the baseline value. The EC_50_ values, fixed at 50% of the total leakage, were calculated from a variable-slope sigmoidal dose–response curve for the leakage percentage as a function of the sample concentration (log10). Melittin, a benchmark cytolytic peptide from bee venom, was used for comparison. Melittin was purified from the apitoxin of apiary Carniolan hybrid honeybee (*Apis mellifera* L.) collected by the Department of Bee Research, Plant Protection Research Institute, Agricultural Research Centre, Egypt.

## Figures and Tables

**Figure 1 toxins-16-00209-f001:**
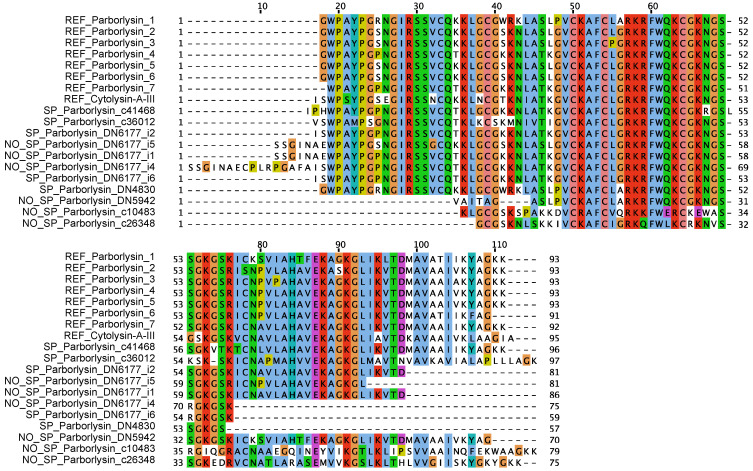
Alignment of parborlysin-like sequences found in the transcriptomes. N-terminal sequence boundaries were predicted using SignalP-5.0. Sequence names: REF; reference sequence, SP; SignalP-5.0-processed, NO_SP; no signal sequence found. Sequence identifiers ending in _cXXX were found in the transcriptome sequenced in this study, and identifiers ending in _DNXXX were found in SRX731467. Default Clustal colour code is used.

**Figure 2 toxins-16-00209-f002:**
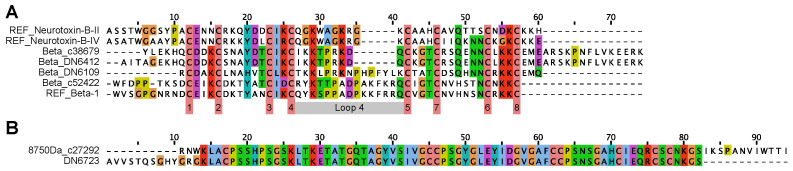
ClustalW alignment of conoprec-processed beta/neurotoxin-B and 8750 Da sequences found in the *P. corrugatus* transcriptomes. (**A**) Beta/neurotoxin B hit alignment. REF; neurotoxins B II and IV on top; nemertide beta 1 at the bottom. Grey bar marks loop 4; numerals mark Cys residues. (**B**) ClustalW alignment of conoprec-predicted mature 8750Da BLASTp hits. DN6723 does not contain a signal sequence and seems to be truncated after position 82.

**Figure 3 toxins-16-00209-f003:**
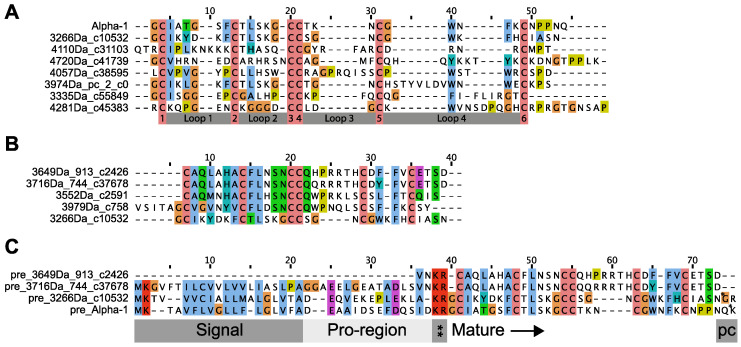
Transcriptome mining for alpha nemertides and the 3624 Da peptide. (**A**) Alignment of conoprec-predicted mature hits with similarity to alpha nemertides. Alpha 1 on top for reference. Cystine and loop numbering are marked. (**B**) Conoprec-predicted mature toxin hits with similarity to 3624 Da peptide. Note that 3266Da_c10532 is found in both the alpha and 3624 surveys. (**C**) Precursor sequences with conoprec-identified regions for 3694, 3716, and 3266 Da peptides. Alpha 1 precursor for reference. PC: post-sequence cleavage site. * Specific post-sequence cleavage sites for 3266 and alpha 1, respectively. ** Pre-sequence cleavage site. Deafault Clustal color code is used.

**Figure 4 toxins-16-00209-f004:**
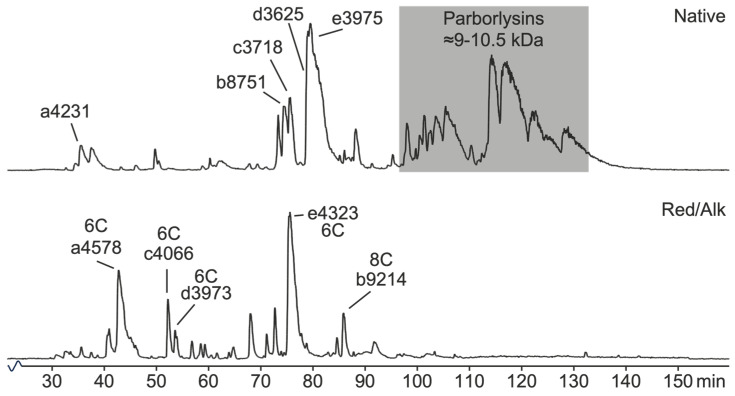
Initial screening of re-suspended lyophilised mucus, and reduced and alkylated sample. Top: native re-suspended mucus. The probable parborlysins (9–10.5 kDa) are marked with a grey box. Bottom: MS chromatogram of gel-filtrated reduced and alkylated mucus. Numbers: deconvoluted average masses to 1 z. Lower-case letters denotate the matching pairs of native and alkylated peptides with an even difference divided by 58 (carbamidomethyl-Cys). #C denotates the number of Cys present.

**Figure 5 toxins-16-00209-f005:**
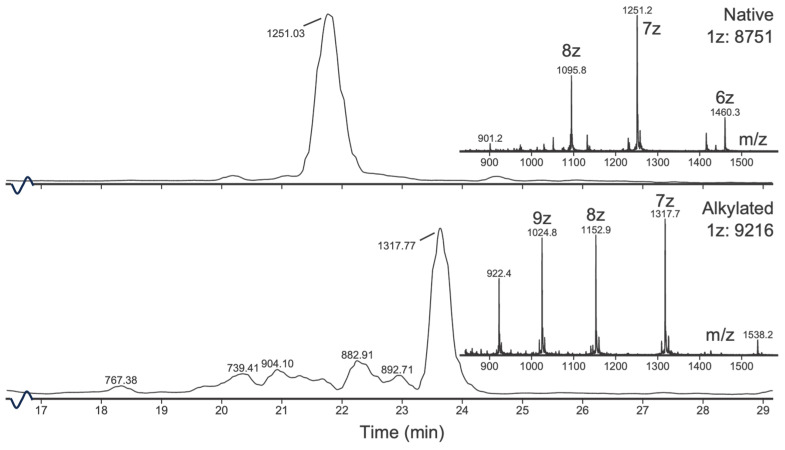
UPLC-QToF chromatogram of native (**top**) and alkylated (**bottom**) 8751 peptides. The increase in mass (464 u) corresponds to eight alkylated Cys residues. Charge states are marked with z in the spectrum. The time axis is truncated (16–29 min).

**Figure 6 toxins-16-00209-f006:**
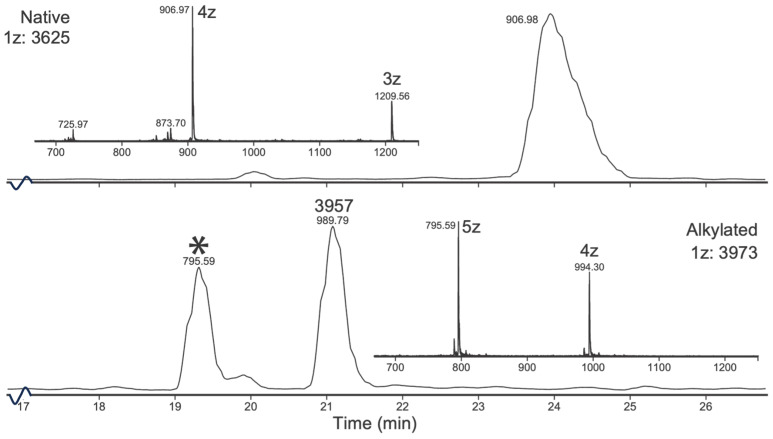
Reduction and alkylation of 3625. UPLC-QToF chromatogram of native (**top**) and alkylated (**bottom**) 3625 peptide. The increase in mass (348 u) corresponds to six alkylated Cys residues. The peak at 21 min (marked with *, 3957 1 z, 989.79 4 z) differs by 16 to the six Cys peptide and cannot be explained by alkylation (no impurity from the native 3624 fraction can explain the dual peaks in the alkylated sample). Charge states are marked with z. The time axis is truncated (16–29 min).

**Figure 7 toxins-16-00209-f007:**
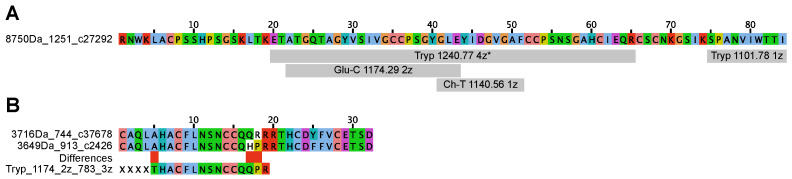
Enzymatic cleavage and tandem MS sequencing of 8750 Da and 3624 Da peptides. (**A**) Conoprec-predicted mature sequence of 8750 Da peptide. Grey boxes represent tandem-MS-sequenced fragments. Tryp: Trypsin, Ch-T: chymotrypsin and Glu-C. * not fully sequenced fragment. (**B**) Conoprec-predicted 3716 Da and 3649 Da peptides aligned with the tryptic fragment 1174 derived from the 3624 Da peptide. The Leu residue is inferred from the alignment. Differences are highlighted in empty red boxes. XXXX marks unsequenced amino acids of a combined mass of 557 Da (b-ion series). Default Clustal colour code is used.

**Figure 8 toxins-16-00209-f008:**
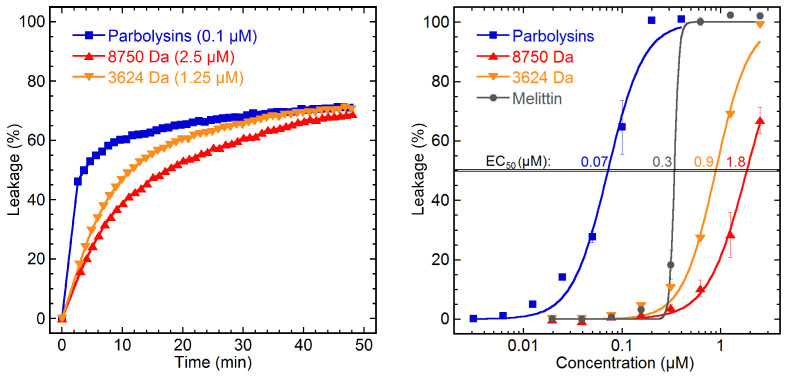
Membrane permeabilization induced by *P. corrugatus* peptides measured by the escape of trapped carboxyfluorescein from *E. coli* phospholipid liposomes. To the left are representative data for liposome leakage as a function of time at each peptide concentration required to reach ≈70% leakage. To the right are leakage levels reached after 45 min as a function of peptide concentration, where each marker represents the mean of four experiments with standard deviations. EC_50_, fixed at 50% of maximum leakage, is highlighted by a horizontal double line with EC_50_ values (µM) indicated for each curve. The 3624 Da curve is derived from a single experiment series.

**Table 1 toxins-16-00209-t001:** Effect of fractions in *Artemia* microwell assay.

Fraction	Total Mass (mg/mL)	Protein Mass (280 nm, mg/mL)	Toxicity (%)
Body	10	6.0	100
	1	0.61	6.25
Mucus ^a^	12.4	0.1	0
LMW	21.4	0.08	0
HMW	0.28	0.18	100
Parborlysins	n/a	1.0	100
	n/a	0.1	7.1
Control	0	0	1.3 ^b^

LMW: low-molecular-weight fraction from mucus. HMW: high-molecular-weight fraction from mucus. n/a: not available. ^a^: lyophilised mucus-containing water. ^b^: n = 75, 12 wells. Protein mass is based on average extinction coefficients from previously described parborlysins [13]. All experiments were performed in duplicate.

**Table 2 toxins-16-00209-t002:** Summary of fractions and peptides used in the membrane permeabilization assay.

ID	MW (Da)	nr aa	rt (min)	Net Charge at pH 7 *
Parborlysins	9.5 k–10.5 k	88–97 ^a^	29	16.1 to 19.1 ^b^
8750 Da	8750	84	22	3.2
3624 Da	3624	32	24	−0.8 ^c^
Melittin	2845	26	n/a	5

MW: monoisotopic molar mass. nr aa: number of amino acid residues in the sequence. rt: retention time in RF-HPLC. n/a: not available. *: data calculated at pepcalc.com (innovagen). ^a^: approximated from the MW by the average amino acid residue MW (108 Da). ^b^: calculated from the reference parborlysin sequences, including the two found in this study. ^c^: calculated using the tryptic fragment from tandem MS sequencing for 3624, and the C-terminal of 3649.

## Data Availability

The data presented in this study are available upon request from the corresponding author.

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
