# Peer review of "Peptide Toxins from Antarctica: The Nemertean Predator and Scavenger Parborlasia corrugatus (McIntosh, 1876)"

_toxins, 2024, doi:10.3390/toxins16050209_

Round 1
Reviewer 1 Report
Comments and Suggestions for Authors
In the manuscript (ID toxins-2905511), authors researched peptide toxins from the top predators -- nemertean worm Parborlasia corrugatus (McIntosh, 1876) in Antarctic waters. In general, this topic has scientific significance for us to understand the peptide toxins from marine invertebrates and the manuscript has readability. In addition, the research content also meets the content requirements of Toxins. Therefore, I think the manuscript can be accepted and published in Toxins after a minor revision. To ensure the publication readiness of the manuscript, the following issue needs to be addressed:
(1) Abstract:
Line 14-15: Suggest the author rewrite this sentence. In addition, it is suggested that the author can give some key experimental results in the abstract, which is more convincing and readable。
(2) Key Words
--Line 18: Should be “nemertean worm Parborlasia corrugatus” rather “Nemertea”.
(3) Introduction: The introduction provides an adequate background on the topic.
--Line 31: Should be “remote area -- the Antarctic” rather “remote area; the Antarctic”.
--Line 60: Should be “have been investigated” rather “has been investigated”.
(4) Results
--Line 137: MSMS. When the full name appears in the preceding manuscript, the abbreviation is written after the full name, followed by the abbreviated form. This abbreviation can be written directly in the manuscript.
--Line 119: Should be “1 mg/mL” rather “1mg/mL”.
--Data analysis, especially significance analysis, should be applied to all experimental results of the manuscript, such as Table 1.
(5) Conclusion
--It is suggested that the author make an explanation for the subsequent research of this manuscript.
(6) Materials and Methods
--It is suggested that the author make a general flow chart of separation, identification and activity determination according to the content of this manuscript, which is more conducive to readers' reading and understanding of this experiment.
--Line 474 and 482: anti-microbial peptide (AMP), high molecular weight (HMW) fraction and a low molecular weight (LMW). Authors should explain these abbreviations when they first appear.
(7) References
--It is recommended that the author carefully check the references and unify the format of the references according to the requirements of the journal, such as
--Line 571 and 574: Should be “Parborlasia corrugatus” rather “Parborlasia Corrugatus”. In addition, “Parborlasia corrugatus” should be in italics. In addition, it is recommended that the authors also check other parts of the manuscript for similar problems.
Comments on the Quality of English LanguageMinor editing of English language required.
Author Response
Reviewer 1
In the manuscript (ID toxins-2905511), authors researched peptide toxins from the top predators -- nemertean worm Parborlasia corrugatus (McIntosh, 1876) in Antarctic waters. In general, this topic has scientific significance for us to understand the peptide toxins from marine invertebrates and the manuscript has readability. In addition, the research content also meets the content requirements of Toxins. Therefore, I think the manuscript can be accepted and published in Toxins after a minor revision. To ensure the publication readiness of the manuscript, the following issue needs to be addressed:
(1) Abstract: Line 14-15: Suggest the author rewrite this sentence. In addition, it is suggested that the author can give some key experimental results in the abstract, which is more convincing and readable
The abstract has been rewritten in lines 14-15, according to the reviewer’s suggestion.
(2) Key Words--Line 18: Should be “nemertean worm Parborlasia corrugatus” rather “Nemertea”.
Has been changed according to the reviewer’s suggestion.
(3) Introduction: The introduction provides an adequate background on the topic.
--Line 31: Should be “remote area -- the Antarctic” rather “remote area; the Antarctic”.
--Line 60: Should be “have been investigated” rather “has been investigated”.
All suggested changes implemented.
(4) Results
--Line 137: MSMS. When the full name appears in the preceding manuscript, the abbreviation is written after the full name, followed by the abbreviated form. This abbreviation can be written directly in the manuscript.
The abbreviation has now been defined in the text.
--Line 119: Should be “1 mg/mL” rather “1mg/mL”.
Change implemented.
--Data analysis, especially significance analysis, should be applied to all experimental results of the manuscript, such as Table 1.
Significance analysis is not attainable for Table 1 as these were performed in duplicates, which we believe should be clear from legend and materials & methods section. However, we have tried to clarify this somewhat in the materials & methods section. For figure 8 (where standard deviation is shown) we have appended the original data in xls format as Supplementary material.
(5) Conclusion
--It is suggested that the author make an explanation for the subsequent research of this manuscript.
We have attempted to elaborate on this in the final conclusion segment.
(6) Materials and Methods
--It is suggested that the author make a general flow chart of separation, identification and activity determination according to the content of this manuscript, which is more conducive to readers' reading and understanding of this experiment.
In the interest of keeping the manuscript succinct (albeit already 16 pages) we have not introduced a flow chart, deemed perhaps not necessary. If the editor so requests, we can of course do this.
--Line 474 and 482: anti-microbial peptide (AMP), high molecular weight (HMW) fraction and a low molecular weight (LMW). Authors should explain these abbreviations when they first appear.
The explanations referred to on lines 474 and 482 have now been removed as they are already present earlier in the manuscript.
(7) References
--It is recommended that the author carefully check the references and unify the format of the references according to the requirements of the journal, such as
--Line 571 and 574: Should be “Parborlasia corrugatus” rather “Parborlasia Corrugatus”. In addition, “Parborlasia corrugatus” should be in italics. In addition, it is recommended that the authors also check other parts of the manuscript for similar problems.
This has been done.
Reviewer 2 Report
Comments and Suggestions for Authors
This manuscript claims to have discovered a new type of polypeptide toxins containing 8 cysteines from the nemertean worm. But, the data provided are insufficient.
1. The transcriptome data provided are literal only, and do not support specific results. It is recommended to provide specific analysis.
2. In Figure 1, the polypeptide is only an inferred amino acid sequence, and the cysteine connection is not determined. It is suggested to give the connection method by experiment or molecular calculation.
3. In Figure 2, the same polypeptide is only a primary structure, and the connection mode of 8 cysteines has not been determined, which does not support a new class of polypeptides. You are advised to specify the connection mode.
4. In Figure 3, the same polypeptide is only a primary structure, and the linking mode of cysteine is not determined. It is suggested to give the connection method by experiment or molecular calculation.
5. In Figure 4, the horizontal coordinate in the figure uses a string of ICONS, whose meaning is unclear. The advice is clear.
6. Figure 5, should not be the original graph. It is recommended to provide the original map.
7. Figure 6, again, should not be the original graph. It is recommended to provide the original map.
8. In Figure 7, the sequence alone does not support the result. It is recommended to attach the typical original MS spectrum.
9. Figure 8 is a diagram of data processing. The figure with typical experimental results is suggested as Figure A.
To sum up, it is suggested that the manuscript should be revised significantly.
Author Response
Reviewer 2
- The transcriptome data provided are literal only, and do not support specific results. It is recommended to provide specific analysis.
We are not exactly sure about what reviewer 2 means by specific analysis here. We do not overstate any claims, but simply account for the sequences found in the analysis. We would need a clarification on what reviewer 2 actually wants here.
- In Figure 1, the polypeptide is only an inferred amino acid sequence, and the cysteine connection is not determined. It is suggested to give the connection method by experiment or molecular calculation.
We agree that a certain connectivity assessment would strengthen these results, but we are unable to carry this out within the frame of the present project. We do not advocate reporting a “virtual” calculated connectivity, as an erroneous virtual structure has been reported previously: (DOI: 10.1016/j.toxicon.2015.09.044). Awaiting connectivity assessment as suggested by the reviewer, we believe it’s more conducive to abstain at this point.
- In Figure 2, the same polypeptide is only a primary structure, and the connection mode of 8 cysteines has not been determined, which does not support a new class of polypeptides. You are advised to specify the connection mode.
See answer to point 2.
- In Figure 3, the same polypeptide is only a primary structure, and the linking mode of cysteine is not determined. It is suggested to give the connection method by experiment or molecular calculation.
See answer to point 2.
- In Figure 4, the horizontal coordinate in the figure uses a string of ICONS, whose meaning is unclear. The advice is clear.
Has been corrected.
- Figure 5, should not be the original graph. It is recommended to provide the original map.
We have modified all chromatograms so that it’s clearer that the X-axis has been truncated.
- Figure 6, again, should not be the original graph. It is recommended to provide the original map.
Again, we have modified the graph so that it’s clearer that the X-axis has been truncated.
- In Figure 7, the sequence alone does not support the result. It is recommended to attach the typical original MS spectrum.
The MSMS spectrum of tryptic fragment 1101.78 has been added as a Supplementary result.
- Figure 8 is a diagram of data processing. The figure with typical experimental results is suggested as Figure A.
We have modified figure 8 accordingly, and added explanatory text in the legend and on page 8 and 11.
We thank reviewer 2 for thorough work and many good suggestions and trust that the revised manuscript is found acceptable in its entirety.
Reviewer 3 Report
Comments and Suggestions for Authors
The paper reports a suite of predicted nemertides found at the transcriptomic level from the Antarctic top predator, the nemertean worm Parborlasia corrugatus. The transcriptome mining revealed ten putative toxins with a cysteine pattern similar to that of alpha nemertides, four nemertide beta type of sequences, and tow novel full length parborlysins. Nemertean worms express toxins in the epidermal mucus, and the expression of these was determined by LC-MS. The findings include a new type of nemertide with a mass of 8750 Da, containing 84 amino acids, of which eight are cysteines. In addition, the authors report the presence of six cysteine-containing peptides. The toxicity of tissue extracts and fractions from mucus were tested in an Artemia assay, revealing activity in tissue and mucus high molecular weight fraction, in addition to a parborlysin fraction. Membrane permeability experiments display membranolytic activity of some peptides, most prominently the parborlysin fraction with an estimated EC50 of 50 nM. I am not an expert in this field, but I think the results reported in this paper are of great interest in toxins and peptides. Therefore, I recommend the publication in toxins.
Author Response
Reviewer 3
The paper reports a suite of predicted nemertides found at the transcriptomic level from the Antarctic top predator, the nemertean worm Parborlasia corrugatus. The transcriptome mining revealed ten putative toxins with a cysteine pattern similar to that of alpha nemertides, four nemertide beta type of sequences, and tow novel full length parborlysins. Nemertean worms express toxins in the epidermal mucus, and the expression of these was determined by LC-MS. The findings include a new type of nemertide with a mass of 8750 Da, containing 84 amino acids, of which eight are cysteines. In addition, the authors report the presence of six cysteine-containing peptides. The toxicity of tissue extracts and fractions from mucus were tested in an Artemia assay, revealing activity in tissue and mucus high molecular weight fraction, in addition to a parborlysin fraction. Membrane permeability experiments display membranolytic activity of some peptides, most prominently the parborlysin fraction with an estimated EC50 of 50 nM. I am not an expert in this field, but I think the results reported in this paper are of great interest in toxins and peptides. Therefore, I recommend the publication in toxins.
No action was required from Reviewer 3.
Round 2
Reviewer 2 Report
Comments and Suggestions for Authors
1. Line 255, Figure 1 should be S1.
2. Line 297, there should be a space between E. coli and phospholipid.
Author Response
1. Line 255, Figure 1 should be S1.
2. Line 297, there should be a space between E. coli and phospholipid.
The changes have been done.